# Learning Disentangled Representations and Group Structure of Dynamical Environments

**Robin Quessard**[1,2*]    **Thomas D. Barrett**[3*]    **William R. Clements**[1*]
[1]Indust.ai, Paris, France
[2]École Normale Supérieure, Paris, France
[3]University of Oxford, Oxford, UK
robin.quessard@ens.fr, thomas.barrett@physics.ox.ac.uk, william.clements@indust.ai

## Abstract

Learning disentangled representations is a key step towards effectively discovering and modelling the underlying structure of environments. In the natural sciences, physics has found great success by describing the universe in terms of symmetry preserving transformations. Inspired by this formalism, we propose a framework, built upon the theory of group representation, for learning representations of a dynamical environment structured around the transformations that generate its evolution. Experimentally, we learn the structure of explicitly symmetric environments without supervision from observational data generated by sequential interactions. We further introduce an intuitive disentanglement regularisation to ensure the interpretability of the learnt representations. We show that our method enables accurate long-horizon predictions, and demonstrate a correlation between the quality of predictions and disentanglement in the latent space.

## 1 Introduction

Representation learning occupies a central place in the machine learning literature (Bengio et al. (2013); Ridgeway (2016)). A good representation should ideally reflect and disentangle the underlying data generation mechanisms, and can be used to efficiently predict, classify, or generalize. For example, a robot learning to grasp a variety of objects would benefit from having access to a representation that distinguishes between scene properties such as object color, size, or orientation (Suter et al. (2019)). However, learning interpretable representations of data that explicitly disentangle the underlying mechanisms structuring observational data is still a challenge.

To address this challenge, one can begin by drawing a parallel between the pursuit of underlying structure in machine learning and in physics. In machine learning, data is structured by its underlying generative factors, which can be understood as the degrees of freedom that, when changed, independently and tractably modify the generated data. Physics often searches for structure using group representation theory by considering the infinitesimal transformations that generate the symmetry group of a physical environment (Lie (1893); Weinberg (1995)). In both physics and machine learning, one has to find a faithful – and, ideally, interpretable – representation of these generative factors to structure the representation one has of the environment. This connection between representation learning in machine learning and representations in physics was previously highlighted in Higgins et al. (2018). However, although Higgins et al. (2018) formalise a definition of disentangled representations by analogy to physics, if and how such disentangled representations can be learnt from environmental observations remains an open question.

---

[*]All authors contributed equally

In this work, we address these challenges and extend the work of Higgins et al. (2018) by proposing a method for learning disentangled representations of dynamical environments from a dataset of past interactions. Our method focuses on learning the structure of the symmetry group ruling the environment's transformations, where symmetry transformations are understood as transformations that do not change the nature of the objects in the environment. For this purpose, we represent dynamical environments by encoding observations as elements of a latent space and representing transformations as special orthogonal matrices that act linearly on the latent space. We also introduce an intuitive regularisation that encourages the disentanglement of the learned representations, and experimentally demonstrate that our method successfully learns the underlying structure of several different explicitly symmetric environments.

## 2   Related work

Different definitions of what constitutes, and how to learn, a disentangled representation have been put forward (Bengio et al. (2013); Higgins et al. (2018); Suter et al. (2019)). Some success has been found in identifying loosely defined generative factors in static datasets (Higgins et al. (2017a); Chen et al. (2016); Karras et al. (2019)), with applications to, for example, domain adaptation (Higgins et al. (2017b)). However, it has been argued that for effective representation learning one should consider not only static data but also the ways in which this data can be transformed (Higgins et al. (2018)) or interacted with (Thomas et al. (2017)).

Specifically, Higgins et al. (2018) propose a definition of disentangled representations based on the physical notion of group representations of symmetry transformations. These representations cannot be learned without some form of interaction with the environment (Caselles-Dupré et al. (2019)). However, Higgins et al. (2018) did not propose a specific method for learning such representations. Prior work on learning group structure of latent space representations from data typically assumes prior knowledge of the symmetry group (Jaegle et al. (2018); Cohen & Welling (2014a,b); Caselles-Dupré et al. (2019)). Connor & Rozell (2020) do not assume prior knowledge of the symmetry group, however their method for learning the transformations requires that the observations are already encoded into a structured latent space. They also do not provide a method for enforcing disentanglement. To the best of our knowledge, our work is the first to learn the underlying group structure of environments and disentangled representations (as defined in Higgins et al. (2018)) without any prior knowledge of the symmetry group.

Parallels can be drawn between our work and state-space models in machine learning, which consider representations of both observations and the dynamics that act linearly on the latent space (Watter et al. (2015); Karl et al. (2017); Fraccaro et al. (2017); Miladinović et al. (2019); Li et al. (2019)). However, these methods do not reveal the underlying structure of the transformations. Moreover, it is not straightforward to reconcile the state-space model objective of modelling probabilistic generative processes with the inherently deterministic framework of Higgins et al. (2018).

In parallel, physics and machine learning have been forging strong ties based for example on Hamiltonian theory and the integration of ordinary differential equations in the latent space to describe the evolution of dynamical systems (Chen et al. (2018); Toth et al. (2019); Greydanus et al. (2019)). Finally, Hamiltonian-based methods have also been used to discover specific symmetries of physical systems (Bondesan & Lamacraft (2019)).

## 3   Disentangled representations

In this section, we review the notion of symmetry-based disentangled representations, which is based upon the definition provided by Higgins et al. (2018). For a more detailed discussion, or for readers unfamiliar with group theory and symmetry groups, we refer to the original work of Higgins et al. (2018).

We consider both an environment that returns observations and a set of transformations that act on this environment. As in physics, we assume that those transformations are elements of a symmetry group $G$. For example, transformations that rotate a 3D object are elements of the $SO(3)$ group of rotations. Informally, we will have learnt a representation of this environment if we can map observations of the environment to elements of a latent space and map symmetry transformations to

linear transformations on this latent space such that the underlying structure of the environment is preserved.

Formally, a representation of an environment as defined in Higgins et al. (2018) maps an observation space $X$ to a latent space $V$ with $f : X \to V$, and maps symmetry group $G$ to a linear representation (in the algebraic sense) on $V$ (Hall (2015)). This means finding a homomorphism $\rho : G \to GL(V)$ between the symmetry group $G$ and the general linear group of the latent space $GL(V)$ so that the map in Figure 1 is equivariant. For example, if we consider an observation $x_1$ of a 3D object and a transformation $g$ that rotates the object, the new observation after the transformation is $x_2 = g.x_1$. We need to ensure $f(g.x_1) = \rho(g).f(x_1)$. We make the common shortcut of omitting the notation $\rho(g)$ and we use $g$ to denote the transformation in both the observation space and the latent space.

Another requirement we wish to make concerning the representation to be learned is that it is disentangled in the sense of Higgins et al. (2018). Formally, if there exists a subgroup decomposition of $G$ such that $G = G_1 \times G_2... \times G_n$, we would like to decompose the representation $(\rho, V)$ in subrepresentations $V = V_1 \oplus V_2... \oplus V_n$ such that the restricted subrepresentations $(\rho_{|G_i}, V_i)_i$ are non-trivial and the restricted subrepresentations $(\rho_{|G_i}, V_j)_{j \neq i}$ are trivial (we recall that a trivial representation of $G$ is equal to the identity for every element of the group $G$).

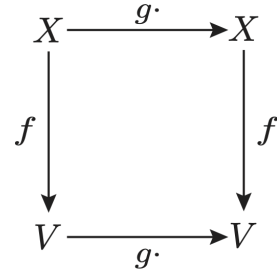

Figure 1: Equivariant map between the observation space $X$ and the latent space $V$.

This definition of disentangled representations has several advantages. First, it maps onto an intuitive notion of disentangled representation as one that separates the data generative factors into different subspaces. It also provides a principled resolution to several points of contention concerning what should be considered a data generative factor, which ones can be disentangled and which ones cannot, and what dimensionality the representation of each factor should have. However, despite theoretical analysis by Higgins et al. (2018) and Caselles-Dupré et al. (2019), to the best of our knowledge no practical method for learning such representations has been put forward.

## 4 Methods

We consider a dataset of trajectories $(o_0, g_0, o_1, g_1, ...)$, each consisting of sequences of observations $o$ and transformations $g$ that generate the next observation. We interpret each transformation as an element of a symmetry group of the environment. Our goal is to map observations to a vector space $V$ and interactions to elements of $GL(V)$ to obtain a disentangled representation of the environment as defined in the previous section, with no prior knowledge of the actual symmetries.

### 4.1 Parameterisation

Here, we propose a method for mapping interactions $g_i$ to elements of $GL(V)$. We consider matrix representations of $GL(V)$. Since we are looking for a matrix representation of an *a priori* unknown symmetry group $G$, we need to consider a group of matrices with a learnable parameterisation large enough to potentially contain a subgroup that is a representation of $G$.

We propose to represent $G$ by a group of matrices belonging to $SO(n)$, which is the set of orthogonal $n \times n$ matrices with unit determinant. Given its prevalence in physics and in the natural world we can expect $SO(n)$ to be broadly expressive of the type of symmetries we are most likely to want to learn. As orthogonal matrices conserve the norm of vectors they act on, we will consider observations encoded in a unit-norm spherical latent space, as was done for example by Davidson et al. (2018) in the context of VAEs.

We parameterise the $n$-dimensional representation of any transformation $g$ as the product of $n(n-1)/2$ rotations (Pinchon & Siohan (2016); Clements et al. (2016))

$$g(\theta_{1,2}, \theta_{1,3}.., \theta_{n-1,n}) = \prod_{i=1}^{n-1} \prod_{j=i+1}^{n} R_{i,j}(\theta_{i,j}) \tag{1}$$

where $R_{i,j}$ denotes the rotation in the $i, j$ plane embedded in the $n$-dimensional representation. For example, a 3-dimensional representation has three learnable parameters, $g = g(\theta_{1,2}, \theta_{1,3}, \theta_{2,3})$, each parameterising a single rotation, such as

$$R_{1,3}(\theta_{1,3}) = \begin{pmatrix} \cos\theta_{1,3} & 0 & \sin\theta_{1,3} \\ 0 & 1 & 0 \\ -\sin\theta_{1,3} & 0 & \cos\theta_{1,3} \end{pmatrix}. \tag{2}$$

These parameters, $\theta_{i,j}$, are learnt jointly with the parameters of an encoder $f_\phi$ mapping the observations to the $n$-dimensional latent space $V$ and a decoder $d_\psi$ mapping the latent space to observations (see supplementary information for pseudo-code). The training procedure consists of starting from a randomly selected observation $o_i$ within the dataset of trajectories, encoding it with $f_\phi$, and then transforming the latent vector $f_\phi(o_i)$ using the representation matrices of the next $m$ interactions in the dataset $\{g_k\}_{k=i+1,\dots,i+m}$. The result of those linear transformations in the latent space is decoded with $d_\psi$ and yields $\hat{o}_{i+m}$,

$$\hat{o}_{i+m}(\phi, \psi, \theta) = d_\psi(g_{i+m}(\theta).g_{i+m-1}(\theta)....g_{i+1}(\theta).f_\phi(o_i)) \tag{3}$$

The training objective is the minimization of the reconstruction loss $\mathcal{L}_{\text{rec}}(\phi, \psi, \theta)$ between the true observations $\{o_k\}_{k=i+1,\dots,i+m}$ obtained after the successive transformations in the environment and the reconstructed observations $\{\hat{o}_k\}_{k=i+1,\dots,i+m}$ obtained after the successive linear transformations using the representations $g_k(\theta)$ on the latent space.

## 4.2 Disentanglement

As explained in section 3, for a representation to be disentangled, each subgroup of the symmetry group should act on a specific subspace of the latent space. We want to impose, without supervision, this disentanglement constraint on the set of transformations $\{g_a\}_a$ that act on the environment. In order to do so without any prior knowledge of the structure of the symmetry group, our intuition is that if each transformation $g_a$ acts on a minimum of dimensions of the latent space, then the representation is disentangled.

We formalize this notion into a metric $\mathcal{L}_{\text{ent}}$ proper to our parameterisation. We choose a metric that quantifies sparsity and interpretability through the number of rotations (each of which is parameterised by $\theta_{i,j}^a$) involved in the transformation matrices $g_a(\theta_{i,j}^a)$. The smallest non-trivial transformation matrix involves a single rotation, so $\mathcal{L}_{\text{ent}}$ measures the use of any additional rotations :

$$\mathcal{L}_{\text{ent}}(\theta) = \sum_a \sum_{(i,j)\neq(\alpha,\beta)} |\theta_{i,j}^a|^2 \qquad \text{with} \quad \theta_{\alpha,\beta}^a = \max_{i,j}(|\theta_{i,j}^a|) \tag{4}$$

The higher $\mathcal{L}_{\text{ent}}$, the higher the entanglement of the representation of the set of transformations. Minimizing this metric $\mathcal{L}_{\text{ent}}$ makes sure that for each transformation $g_a \in \{g_a\}_a$, most of the parameters appearing in the representation of this transformation go to 0, which implies that the transformation acts on a minimum of dimensions of the latent space. If there is only one non-zero parameter in the parameterisation of a transformation, then it only acts on 2 dimensions of the latent space.

## 5 Experiments

In this section, we experimentally evaluate our method for learning disentangled representations in several explicitly symmetric environments with different types of symmetries. The code to reproduce these experiments is provided in notebook form at `https://github.com/IndustAI/learning-group-structure`.

## 5.1 Learning the latent structure of a torus-world

Our first goal is to show that the parameterisation and the training method described in 4.1 allows us to extract useful information about the structure of an environment from inspecting the topology of the learnt latent space. We consider a simple environment similar to that studied in Higgins et al. (2018) and Caselles-Dupré et al. (2019), consisting of pixel observations of a ball evolving in a 2-dimensional space. At each step, the ball moves one discrete step left, right, up, or down, in such a way that the ball's position is confined to a $p \times p$ grid. We impose periodic boundary conditions,

so for example if the ball leaves the grid at the top it reappears at the bottom. We implement this environment using Flatland (Caselles-Dupré et al. (2018)), consider at first $p = 10$, and collect data using a random policy.

A 2-dimensional plane with periodic boundary conditions is topologically equivalent to a torus, and it is this topology that we aim to learn from the dynamics of the environment. Concretely, the symmetry group of this environment is the finite group $G = C_p \times C_p$ where $C_p$ denotes the cyclic group of order $p$ (also called $\mathbb{Z}/p\mathbb{Z}$ or $\mathbb{Z}_p$) and is a finite subgroup of $SO(2) \times SO(2)$. In order to learn a representation of this environment, we need to learn an encoder, a decoder and the representation matrices for the 4 transformations $g_{\text{up}}, g_{\text{down}}, g_{\text{left}}$ and $g_{\text{right}}$ that generate $G$.

To learn this group structure from data, our only assumption is that it can be represented with orthogonal matrices $g \in SO(n)$. We note that this is a weaker assumption than that made in Caselles-Dupré et al. (2019) for this environment, where the matrices were assumed to be identity except for a single $2 \times 2$ block along the diagonal. We will use $n \geq 4$; as discussed in Caselles-Dupré et al. (2019), real representations of cyclic groups can be seen as rotations in planes, and since the symmetry group consists of the direct product of 2 cyclic groups we minimally need 2 planes so 4 dimensions. We start with $n = 4$, and recall that a matrix of $SO(n)$ has $n(n - 1)/2$ degrees of freedom so the matrices of this 4-dimensional representation each have 6 parameters.

We use convolutional neural networks for the encoder and the decoder. The encoder $f_\phi$ has normalized outputs so that it always maps observations to unit-norm latent vectors. We learn jointly the encoder parameters $\phi$, the decoder parameters $\psi$ and the $6 \times 4 = 24$ parameters of the 4 transformation matrices $g_{\text{up}}, g_{\text{down}}, g_{\text{left}}$ and $g_{\text{right}}$ which we denote as $\theta$.

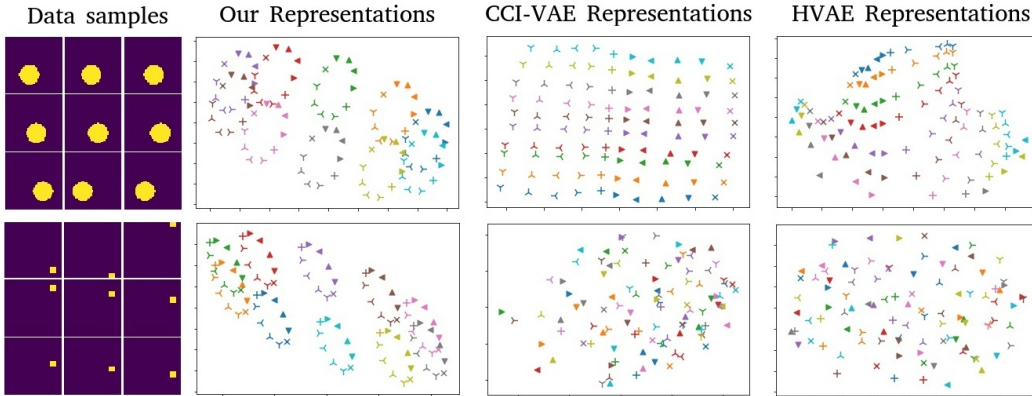

Figure 2: Left column: sample sequences of observations for two different environments. Other columns: 2D projections of the 4D representations of equally spaced observations from these environments, learned by our method, a CCI-VAE, and a hyperspherical VAE. Each symbol/colour pair corresponds to a unique state of the environment; this mapping is consistent for all latent spaces. Top row: environment that returns pixel observations of a ball on a 2D plane with periodic boundary conditions. The underlying structure of this environment is toroidal. Bottom row: a similar environment that returns one-hot observations encoding the position of the ball. Since observations no longer overlap, here VAE-based methods fail to find any structure from observations alone.

Our results are shown in Figure 2 (top row) in which we see that our method successfully learns a 4-dimensional representation of the finite symmetry group $C_p \times C_p$ for $p = 10$. The explicit toroidal structure of the latent space respects the structure of the symmetry group. We compare these representations with those learnt by two other representation learning methods that are based on static observation data: CCI-VAE (Burgess et al. (2017)), which is a state of the art disentangling method, and hyperspherical VAE (Davidson et al. (2018)), which like our method also uses a spherical latent space. As in Caselles-Dupré et al. (2019), we find that in the four dimensional latent space the CCI-VAE ignores two dimensions and places the observations in a grid, which reflects the observational structure of the environment but not its underlying dynamics. Hyperspherical VAE also learns to place observations in a grid-like structure on a sphere.

To further investigate the difference between the observational structure discovered by VAEs and the dynamical structure discovered by our method, we consider a variation of this environment in which

no structure can be found from observations alone. Instead of pixel observations and convolutional networks for the encoder and decoder, we consider a one-hot encoding of the ball's position on a $p \times p$ grid and fully connected neural networks. Our results are shown in figure 2 (bottom row). We find that our method still correctly identifies the dynamical structure of this environment, whereas neither VAE-based method finds any structure.

## 5.2 Controlling the entanglement of the representation

Having found that our method successfully learns a meaningful representation of this environment, now we turn to investigating the disentanglement of this representation using the metric introduced in 4.2. Indeed, many 4-dimensional representations of $C_p \times C_p$ exist, and most of them are entangled. Since the transformation matrices are parameterised as products of 6 rotations, if most of the 6 parameters are non-zero, then the transformation is poorly interpretable because all dimensions of the latent space are mixed after acting on it with this representation.

Using the entanglement metric from section 4 as a regularisation term, we are able to control the entanglement of the learnt representation. Figure 3 compares the learnt transformations between a regularisation minimising the entanglement (figure 3a) and the absence of such regularisation (figure 3b). Even though both representations encode $C_{10} \times C_{10}$ and exhibit the corresponding toroidal structure, the maximally disentangled representation is much more interpretable. The up/down transformations rotate in a single plane (dimensions 1 and 4) by $\pm\pi/5$, whereas the left/right transformations act equivalently in an orthogonal space (dimensions 2 and 3). This is the most intuitive 4-dimensional disentangled representation of the symmetry group $C_{10} \times C_{10}$ we could have learnt.

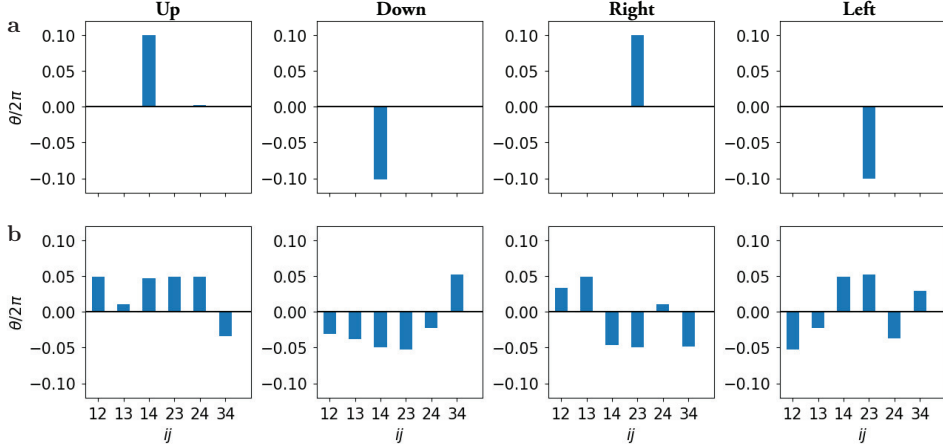

Figure 3: Learnt transformations for a grid-world with $p{=}10$. Values of the angles $\theta_{i,j}/2\pi$ learned for representations (a) with regularisation on the loss to minimise the entanglement, giving $\mathcal{L}_{\text{ent}} = 1.2 \times 10^{-4}$, and (b) without regularisation, $\mathcal{L}_{\text{ent}} = 0.32$.

In the supplementary information, we study the same environment for different values of $n$ (latent space dimension) and $p$ (periodicity of the environment). In particular, we find that when $n > 4$, our method still only learns 4-dimensional representations and disregards superfluous dimensions.

## 5.3 Learning more complex structure

We now investigate whether our method can learn disentangled representations of environments with more complex symmetry group structures. We consider an environment in which observations are pixel observations of a 3D teapot (Crow (1987)), and 8 different interactions. Of these interactions, 6 interactions (denoted $x+$, $x-$, $y+$, $y-$, $z+$ and $z-$) consist of rotating the scene by a fixed $\pm 2\pi/5$ angle around the $x$, $y$, and $z$ axes, and 2 interactions (which we note color+ and color-) involve cycling forwards and backwards through a set of 5 different background colors. Using data collected by a random policy, we wish to disentangle two factors of variation: the spatial rotations and the changes of color.

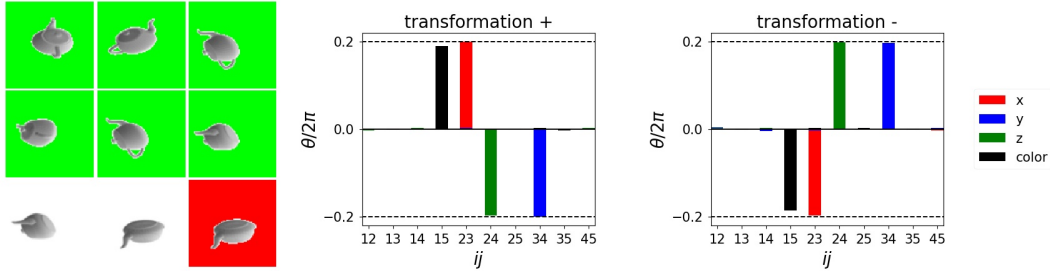

Figure 4: Left: Sequential observations from the environment. Right: Representations learned for the 8 transformations. The 3D rotations of $\pm 2\pi/5$ are associated with dimensions 2, 3 and 4 of the latent space. The cyclic change of color is associated with dimensions 1 and 5 of the latent space.

The symmetry group generated by those transformations, and which we aim to learn, therefore lies in $SO(2)_{\text{color}} \times SO(3)_{\text{rotation}}$. As explained in Higgins et al. (2018), a disentangled representation of 3D rotations should not be able to separate the action of different rotations into distinct latent space dimensions. Indeed, $SO(3)$ cannot be written as a non-trivial direct product of subgroups, therefore, there is no representation in which two rotations around different axes would act on two different subspaces of the latent space. We can still satisfy ourselves with a disentangled representation in which rotations around the x, y and z axes each act on a minimum of dimensions of the latent space, which is a definition of entanglement aligned with the metric we introduced in 4.2.

We choose to learn a 5-dimensional representation of this environment because an interpretable disentangled representation would associate a 3-dimensional subspace to the spatial transformations and a 2-dimensional subspace to the color transformations. Nevertheless, using a higher-dimensional latent space does not change the learnt representation as the disentanglement objective makes unnecessary dimensions impactless and present in none of the transformation matrices.

Our results are shown in figure 4. We find that we effectively learn a 5-dimensional disentangled representation of the environment that conserves the symmetry group structure. Specifically, the 6 interactions that correspond to spatial rotations affect a three-dimensional subspace, which is the minimal subspace in which 3D rotations can be represented. Moreover, all 6 rotation interactions are correctly mapped to rotations of $2\pi/5$. The other 2 colour interactions affect a different two dimensional subspace, which correctly reflects the cyclic nature of the colour changes with a periodicity of 5. We highlight that this is achieved without supervision, as there is nothing to distinguish the 8 interactions in the data apart from their effect on the observations.

## 5.4 Learning disentangled representations of Lie groups

We now show that we are able to learn continuous groups corresponding to infinite sets of transformations and are therefore not limited to environments with discrete actions. We consider pixel observations of a ball evolving on a sphere under the continuous set of rotations around the 3 axes $x, y$ and $z$ in the interval $[-\pi, \pi]$, illustrated in figure 5.

To learn a continuous group of symmetry transformations, also known as a Lie group, we now require a continuous mapping $\rho : G \rightarrow GL(V)$. Whereas in our previous experiments we used a look up table to map each discrete transformation to a set of $\theta$, we now use a neural network $\rho_\sigma$ to learn this mapping for continuous transformations. The network takes as input the transformation in the environment, a concatenation of a scalar denoting the rotation axis (0 for $x$, 1 for $y$ and 2 for $z$) and the value of the angle of the rotation around this axis. It outputs $n(n-1)/2$ scalars parameterising the $n$-dimensional representation of this transformation. In this case, we aim to learn the 3D rotations around the axes of a sphere so we use a 3-dimensional latent space to represent the symmetry group $SO(3)$. For example, the representation of a $\pi/4$ rotation around the $y$ axis is parameterised as a product of 3 rotations, $g_\sigma(1, \pi/4) = R_{1,2}(\theta_{1,2}) \cdot R_{1,3}(\theta_{1,3}) \cdot R_{2,3}(\theta_{2,3})$, where $(\theta_{1,2}, \theta_{1,3}, \theta_{2,3}) = \rho_\sigma(1, \pi/4)$. As previously, the loss function combines a reconstruction loss and an entanglement regularization, which now includes parameters $\sigma$, $\mathcal{L}(\phi, \psi, \sigma) = \mathcal{L}_{\text{rec}}(\phi, \psi, \sigma) + \lambda \mathcal{L}_{\text{ent}}(\sigma)$.

Our results are shown in Figure 5, which show that our method effectively learns a 3-dimensional representation of $SO(3)$, where rotations around each axis act only within a single plane of the

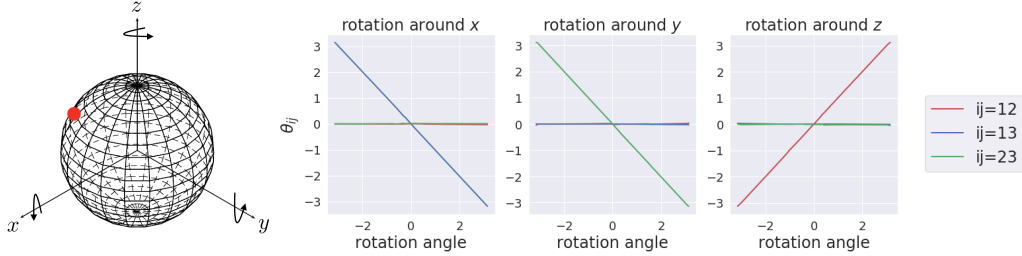

Figure 5: Left: illustration of the environment, in which a ball evolves around a sphere under the action of continuous rotations. Right: Learned values of the parameters $\theta_{1,2}, \theta_{1,3}$ and $\theta_{2,3}$ of the transformation matrices as a function of the angle of the rotation in the environment for each rotation axis $x, y$ and $z$. For any angle and each rotation axis, there is only one non-zero parameter (for example $\theta_{2,3}$ for rotations around the $y$ axis), which makes this representation a well-disentangled and interpretable representation.

latent space. With this 3-dimensional disentangled representation of a continuous group of symmetry transformations in a 3-dimensional space, we have successfully learnt a perfectly interpretable representation of an environment with continuous dynamical transformations.

## 5.5 Multi-step predictions with disentangled representations

A fundamental motivation for, and exhibition of, learning the underlying structure of dynamical environments is to predict the evolution of the system along unseen trajectories. In this section we will examine the quality of long-term predictions produced by our framework, and show the advantage offered by disentangled representations. To do so, we return to the Flatland environment, previously used in sections 5.1 and 5.2, with a 4-dimensional latent space, and consider learning from 10-step trajectories initialised to the centre of a $5\times5$ grid using the same method as in section 5.1.

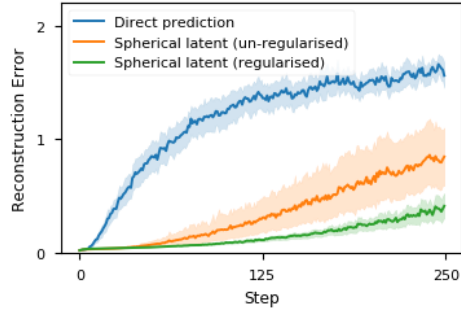

Figure 6: Reconstruction error as a function of the episode length for various models as described in the main text. Shaded areas correspond to 95 % confidence intervals of the mean, measured over 20 training seeds.

We first contrast our approach to a simple "direct prediction" model that does not consider any structure within the environment. Instead, at every time-step, the observation is encoded in a 4D latent space and concatenated with a one-hot encoded action vector, from which the next state is directly decoded. From Figure 6, we see that the performance of "direct prediction" rapidly deteriorates as errors accumulate over many successive encoding and decoding operations. However, our framework – both with and without an additional disentanglement regularisation – demonstrates better predictive performance. Informally, the spherical latent space compactly encodes the relevant environmental structure and the learnt representation is then seen to accumulate errors at a lower rate.

It is also clear from Figure 6 that explicitly disentangled representations provide further improved predictive performance. This is due to the disentanglement regularisation yielding transformation matrices that are closer to a faithful representation of the symmetry group, in that they better conserve important properties of the underlying dynamics such as cyclicity and commutation relations compared to more complex transformation matrices associated with entangled representations. As the transformations learnt with disentanglement regularisation better reflect the underlying dynamics, they also yield more accurate long-term predictions. Our entanglement regularisation can then be considered as an inductive bias towards representations that are likely to exhibit good multi-step generalisation. Ultimately, these results are in agreement with the widespread notion that disentan-

gled representations can be useful for down-stream tasks (Bengio et al. (2013); Ridgeway (2016); Higgins et al. (2017b); Locatello et al. (2019); van Steenkiste et al. (2019)).

# 6    Conclusion

In this work, we have opened the possibility of applying representation theory to the problem of learning disentangled representations of dynamical environments. We have exhibited the faithful, explicit and interpretable structure of the latent representations learnt with this method for explicitly symmetrical environments with a specific parameterisation. By establishing the feasibility of learning physics-based disentangled representations as defined in Higgins et al. (2018), we believe our work constitutes a promising step towards developing representation learning methods that are both principled and scalable.

Our method involves encoding general $n$-dimensional representations as the product of $\mathcal{O}(n^2)$ rotation matrices, which would likely become a significant computational bottleneck for larger models with many generative factors. However, with this choice of parameterisation we are able to clearly demonstrate the key insight that a simple regularisation on the number of latent dimensions on which individual representations act can naturally lead to disentanglement. A natural extension of our work could then be to focus on learning representations of more complex real-world environments, which could require the use of more expressive groups than $SO(n)$ or different parameterisations.

## Broader Impact

We propose a new methodology for learning disentangled representations which is demonstrated on model datasets as a proof of principle, therefore our contribution is not at a stage where it is expected to have immediate societal impact. More broadly, however, improved representations can be expected to underpin powerful ML systems that model real-world environments and data. Interpretability is one of the goals of disentangled representations and does merit particular consideration as it is essentially an attempt to extract the underlying description of the world that the system has learnt. It is therefore important that this interpretation is both fair (free from inherent bias) and presented with sufficient context (for example, the learnt representation could simply be one of many equal valid descriptions of the observed data). In this context, a mathematically rigorous definition of disentanglement, such as that presented by Higgins et al. (2018) and used in this work, would seem to be a good first step, though it cannot be considered to fully address these considerations on its own.

## Funding Disclosure

The authors declare no competing financial interests.

## Acknowledgements

We thank Hugo Caselles-Dupré, David Filliat, Yaël Frégier, Michael Garcia-Ortiz, Irina Higgins, and Sébastien Toth for helpful discussions, and the team at indust.ai for their support. We also thank the Agence pour les Mathématiques en Interaction avec l'Entreprise et la Société (AMIES) for their support.

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
