[Supplementary Material]

# Learning Disentangled Representations and Group Structure of Dynamical Environments - Supplementary Material

## 1 Training Details

### 1.1 Pseudo-Code

---
**Algorithm 1** Pseudo-Code for learning group structure and disentangled representations
---
1: *Inputs:*
2: - Dataset of observed trajectories: $\{(s_0, a_0, s_1), ...\}$
3: - Learnable functions $state\_encoder, state\_decoder$, and $action\_representation$
4: - Hyperparameter $\lambda$ for the regularisation term
5: - Number of consecutive interactions $m$ to consider during training
6: **for** batch of length-$m$ sequences in Dataset **do**
7:    $\mathcal{L}_{\text{rec}} = 0$
8:    **for** sequence in batch **do**
9:       $z_i = state\_encoder(s_i)$
10:       **for** $k$ from $i$ to $i + m - 1$ **do**
11:          $g_k = action\_representation(a_k)$
12:          $z_{k+1} = g_k.z_k$
13:          $o_{k+1} = state\_decoder(z_{k+1})$
14:          $\mathcal{L}_{\text{rec}} \mathrel{+}= BinaryCrossEntropy(o_{k+1}, s_{k+1})$
15:       **end for**
16:    **end for**
17:    $\mathcal{L}_{\text{ent}} = \sum_k \mathcal{L}_{\text{ent}}(g_k)$, where $\mathcal{L}_{\text{ent}}(g_k)$ is calculated using equation 4 in the main text
18:    $\mathcal{L}_{\text{total}} = \mathcal{L}_{\text{rec}} + \lambda \mathcal{L}_{\text{ent}}$
19:    $state\_encoder, state\_decoder, action\_representation \leftarrow Backpropagation(\mathcal{L}_{\text{total}})$
20: **end for**

---

The code to reproduce these experiments is provided in notebook form in the supplementary materials and will be made public.

### 1.2 Description of data

Our Flatland experiments for sections 5.1 and 5.2 consider black and white $84 \times 84$ pixel observations. Each action moves the ball one step up, down, left, or right, in such a way that the position of the ball is confined to a $p \times p$ grid where $p$ is determined by the amplitude of the action. Instead of starting with a static dataset of trajectories we generate a new batch of trajectories on the fly during training, starting from a randomly selected location on the $p \times p$ grid and performing a random sequence of $m = 5$ actions with batches of 16 trajectories.

In the experiment in section 5.3, we consider RGB $84 \times 84 \times 3$ pixel observations of the teapot and its background. The dataset consists of a single trajectory of 1000 consecutive randomly selected actions. For training, we use sequences of length $m = 5$ and batches of 16 trajectories.

For the experiment in section 5.4, we consider $10 \times 10 \times 10$ voxel observations of the location of the ball on a sphere, where the position of the ball is encoded by its density spread over the neighbouring voxels. As for the Flatland experiments, we generate the sequences on the fly starting from a randomly selected location and performing a random sequence of $m = 5$ actions with batches of 16 trajectories. At each step one of the three rotation axes is randomly chosen and an angle selected from a uniform probability distribution within a certain range. We found it helpful to gradually increase this range from $[-\frac{2\pi}{5}, \frac{2\pi}{5}]$ to $[-\pi, \pi]$ over the course of training.

For the experiment in section 5.5, we consider once again the Flatland environment of sections 5.1 and 5.2, except we now use $p = 5$, random sequences of $m = 10$ actions, and all trajectories start at the same central location. With all trajectories starting at the centre, sequences that "loop" around the environment are less common, and whereas our algorithm (regularised or unregularised) still learns the underlying toric structure we found that our regularisation introduces an inductive bias for learning representations that provide better long-term predictions.

## 1.3 Network Architectures and Hyperparameters

Due to the different nature of the observations considered, different neural network architectures were used, which also implies the use of different optimal learning rates and regularisation strengths. We provide an overview of the neural network architectures below; for a more complete description of the neural networks and hyperparameters, we invite the reader to inspect the notebooks in our accompanying code at `https://github.com/IndustAI/learning-group-structure`.

### 1.3.1 Flatland experiments

The encoder consists of a convolutional layer with 5 channels, a kernel size of 10, stride of 3, and ReLU activation, followed by a a multi-layer perceptron with 1 hidden layer with 64 neurons and a ReLU activation. The decoder consists of a multi-layer perceptron with one hidden layer with 64 neurons and ReLU activation, followed by a deconvolutional layer with kernel size 34 and a stride of 10. The Adam optimiser was used to optimize neural network weights and the parameters of the transformation matrices.

### 1.3.2 Teapot experiment

The encoder consists of the same convolutional neural architecure used in Mnih et al. (2015), except that we use half the number of filters in the convolutional layers to reduce computation time. The decoder's architecture is the reverse of the envoder, where deconvolutional layers replace convolutional layers. We use the Adam optimizer.

### 1.3.3 Lie Group experiment

The observations are flattened to a vector before being sent to an encoder, which is a multi-layer perceptron with a single hidden layer of 64 neurons and ReLU activation. The decoder has the same architecture as the encoder (where the input and output dimensions are switched). Network $\rho_\sigma$ also is a multi-layer perception with one hidden layer with 32 neurons and ReLU activation.

In general, we found that the combination of a relatively high learning rate for the Adam optimiser Kingma & Ba (2014) and a large number of parameter updates was helpful in overcoming suboptimal local minima in the loss landscape, which sometimes prevent the reconstruction loss from converging and the underlying dynamics from being correctly learnt. For the experiment with the discrete rotations and color changes on the sphere, we also found it helpful to significantly increase the regularisation loss half-way through training to prevent the rotation angles from sometimes collapsing early in training.

### 1.3.4 3D cars and shapes experiments

These experiments will be presented in section 5. Observations consist of $64 \times 64 \times 3$ RGB images. The Adam optimiser was used to optimize neural network weights and the parameters of the transformation matrices.

The encoder network consists of four convolutional layers (kernel size of 4, stride of 2) with 32, 32, 64 and 64 channels respectively. The convolutional output is then flattened and passed through

a fully connected network with a single hidden layer of 256 neurons. All hidden layers use ReLU activations, with the final output normalised to lie upon the unit sphere in our latent space.

The decoder network first maps the 4 latent values to a 256 dimensional vector using a fully-connected network with a single 256 dimensional hidden layer. This is then reshaped and passed through 4 deconvolution layers (kernel size of 4, stride of 2) with 64, 32, 32 and 3 channels respectively. Again, all hidden layers use ReLU activations with a sigmoid activation applied to the final $64 \times 64 \times 3$ output to map it to valid RGB values.

## 2   2D Projections of the learned representations

Here, we provide further information on how the 2D projections of the learned 4D representations were obtained with our method, a CCI-VAE, and a hyperspherical VAE. For the representations learned with our method, we use a random projection onto a 2D space. Although the toric structure can always be identified no matter which random projection is used, several random projections were attempted until we found a projection that yielded the most visually appealing torus. For CCI-VAE, as in Caselles-Dupré et al. (2019), only 2 dimensions are used to encode the position of the ball; to obtain a 2D projection we selected the two dimensions along which the representations had the highest variance and projected along those. For HVAE, the learned representations tend to resemble a grid projected onto a sphere, which is usually not visually apparent when using different random 2D projections of a 4D latent space. We therefore first projected the representations onto the three dimensions which had the highest variance, and then used a Hammer projection onto 2 dimensions.

## 3   Flatland experiments for different values of $n$ and $p$

Here, we show that our method successfully learns the underlying structure of the environment for other values of $n$ (latent space dimension) and $p$ (periodicity of the environment induced by the discrete actions) than those described in the main text.

First, in figure 1 we show 2D projections of the representations learned for all possible states of the environment for different values of $n$ and $p$, all learned using the same set of hyperparameters. In all cases, the toric structure of the environment is correctly learned.

Next, in figure 2 we show the action representations learned by our method with entanglement regularisation on a Flatland environment with $p = 10$ and different values of $n$. We find that when $n > 4$, our method still learns a compact disentangled representation that only uses 4 dimensions to encode the position of the ball.

## 4   Lie group experiments for different latent space dimensions

Here, we show that our method for learning the structure of transformations described by a Lie group also learns disentangled representations when the latent space has more dimensions than are necessary. Specifically, we repeat the experiment described in section 5.4 in the main text with a latent space dimensions of 4 instead of three. Our results are shown in figure 3. We find that the learned representations still only use three dimensions, which is the minimal number required to represent the transformations.

## 5   Experiments on complex environments

Here, we consider distentangling environments with more complex observations. Specifically, we use established datasets of 3D images of cars Kim & Mnih (2018) and shapes Reed et al. (2015) and in both cases we consider two generative factors with five possible values each (for example, 5 possible colors or orientations). Trajectories through this data are created by randomly sampling transformations of these generative factors, where applying the same transformations five times performs a complete cycle through the five possible values. These experiments have similar structure to the Flatland experiments presented in the main text, but with a symmetry group of $C_5 \times C_5$. The encoder, decoder and the parameters of the transformation matrices were trained using the procedure described in 1.1 with sequence lengths of 20 actions and a batch size of 1. Training

Figure 1: 2D projections of the learned representations for all possible states of the Flatland environment for different values of $n$ (latent space dimension) and $p$ (periodicity of the environment). The toric structure of the environment is correctly learned in all cases.

Figure 2: Action representations learned by our method with regularisation for Flatland with $p = 10$ and $n = 4$ (top), $n = 5$ (centre), and $n = 6$ (bottom). In all cases, only the minimal number of dimensions is effectively used to encode the ball's position information.

Figure 3: Results for our Lie group experiment (see section 5.4 in the main text) with a latent space dimension of 4 instead of 3. We observe that the learned representations still only use 3 dimensions, as desired (dimension 1 is ignored).

consisted of 15000 steps of gradient descent with the entanglement regularisation linearly increased from $\mathcal{L}_{\mathrm{ent}} = 0$ to $\mathcal{L}_{\mathrm{ent}} = 0.1$ over the first 10000 steps.

Figures 4 and 5 show the learnt representations and compare the ground truth and reconstructed images when traversing each generative factor using these. In all cases we see that our formalism successfully disentangles the action representations, whilst still providing faithful reconstructions of the environment. It is noteworthy three of the four generative factors are modified by $2\pi/10$ rotations within a plane. However, this $C_{10}$ symmetry in the latent space can still perfectly represent the underlying $C_5$ symmetry.

Figure 4: Individually varying the underlying generative factors of the 3D shapes dataset. The upper images compares the ground truth and reconstructed observations, with the learnt action representation for stepping one place forwards or backwards in the cycle shown underneath.

Figure 5: Individually varying the underlying generative factors of the 3D cars dataset. The upper images compares the ground truth and reconstructed observations, with the learnt action representation for stepping one place forwards or backwards in the cycle show underneath.