[Reviews · NeurIPS 2020]

Review 1

Summary and Contributions: The paper addresses disentanglement in dynamical systems. The authors operate on the definition of disentanglement popularized by Higgins et al., where a representation is considered disentangled if a group of actions acting upon the representation can be decomposed into nontrivial subgroups---acting upon "slices" of the representation. The authors suggest a learnable implementation of this principle by suggesting to learn the group of SO(n) matrices, along with a regularizer to encourage sparse usage of rotations to ensure rotations. They test on five different simple examples and conclude that they indeed arrive at a more disentangled representation.

Strengths: Motivation: The challenge of disentangled representation is still largely unsolved. Finding practical implementations of the concepts by Higgins et al. is a worthwhile endeavor that deserves attention. As such, the paper tackles an interesting problem. Novelty: To the best of the authors' as well as my knowledge, the concepts of Higgins et al. have not been put into practice. As such, the contribution can be considered interesting. Experiments: The experiments are chosen well and highlight the features of the presented approach. Though limited in their scope, they represent a useful proof of conept.

Weaknesses: Restricted applicability: The paper is not sufficiently clear about the applicability and extensibility of their method. The authors certainly do not claim to "solve" disentanglement. However, I am lacking an explicit discussion just how restrictive their representation is. SO(3) is quite restricted in its expressiveness, even though it quite naturally aligns with our intuition of disentanglement. More so, the representation as a matrix product (general cost of ~O(n^3)) of O(n^2) matrices is worth discussing from a scalability point of view. It is fine to restrict this proof of concept, but it should point out a clearer path forward than the current submission does. Strawman baselines: The baselines are both essentially variants of the variational auto-encoder. While they put more emphasis on the representation and disentanglement than the vanilla VAE, it is unclear what motivated the choice of these baselines. In particular, all chosen data sets are inherently dynamical, which the VAEs explicitly cannot and do not handle. It would have been more interesting to see, e.g., dynamical variants of the VAE (see also the comments below about related work) on these data sets to really judge whether these representations add value. Discussion of determinism and relation to generative models: Both baselines are VAE variants, i.e., probabilistic generative models, for which representation learning has seen piqued interest. The suggested model is a deterministic auto-encoder with a deterministic dynamic component. Under these circumstances, the connections to VAEs and similar models should be made more explicit. Can the suggested architecture be turned into a VAE? Why? Why not?

Correctness: The claims are sound. The presented experiments are largely qualitative, but fine as such.

Clarity: The paper is easy to follow and well-structured.

Relation to Prior Work: I am not sufficiently familiar with the literature on disentangled representations. Keeping that in mind, the related work feels incomplete. In particular, it focuses largely on literature around the approach by Higgins et al. and on work from the past three years. Disentanglement is certainly a more widely studied phenomenon, where a reference to the review paper by Bengio feels insufficient. With regards to learning state-space models and learning dynamics, I am more familiar with the literature. I believe this could be discussed more thoroughly, within and outside related work. Neural learning of SSM dates back to at least [1, 2, 3]. The references mentioned in this submission explicitly mention disentanglement, but the quest for "useful" representations is already present in the aforementioned. As a specific example [2] learns a latent structure from observations that could be considered disentangled according to Higgins et al. As mentioned earlier, a comparison to one of these (or a follow-up) as baselines would have been interesting. [1] https://arxiv.org/abs/1511.05121 [2] https://arxiv.org/abs/1605.06432 [3] https://arxiv.org/abs/1506.07365

Reproducibility: Yes

Additional Feedback: --------- POST-REBUTTAL COMMENTS I have read the rebuttal and the other reviews. I am willing to bump my score by 1 to 5. This reflects that I seem to be too harsh with an idea in a still very nascent field. The reason why it stays below acceptance threshold for me is that I still believe that the paper should be improved before publication. One point I want to stress again: The polynomial nature of the group representation needs to be discussed appropriately. It is a severe bottleneck for this idea to practical fruition. The fact that the individual matrices are sparse does not change that. The product very quickly isn't, which makes this representation very costly.


Review 2

Summary and Contributions: The authors propose a novel approach to learning disentangled representations based on the formalism of symmetry-based disentanglement presented in Higgins et al. (2018). The proposed method is straightforward, yet elegant, and overcomes a key limitation of prior methods that adopt this formalism. In particular, through parametrizing interactions in a learned latent space via a product of rotations (describing SO(n)) and a clever regularizer that encourages latent interactions to be sparse, the proposed method is capable of learning disentangled representations without making strong assumptions about the symmetry-group acting on the environment. The authors evaluate their method on several synthetic datasets of varying complexity. The presented evidence clearly indicates that the proposed method works as intended. However, while these results are generally encouraging and significant in and of their own, more sophisticated datasets that test the limitations of the proposed method are left unexplored.

Strengths: The proposed method is intuitive and relates to recent auto-encoding architectures that have been explored for learning disentangled representations. It can be viewed as a type of recurrent auto-encoder that models a sequence of observations in latent space, but where the transformation on the latent space from one time-step to the next (the recurrent connection) is confined to SO(n) through a sequence of parametrized rotations. Separate parameters are considered for each type of transformation applied (action) in the environment, which can be learned via a reconstruction objective. By implementing the latent interactions in this way disentanglement can be achieved in a clever way. In particular, SO(n) can be decomposed into independent subspaces by penalizing the number of effective rotations used. In particular, by encouraging sparsity among the parameters used in these rotations, their effect is essentially confined to a particular subspace. By augmenting the reconstruction objective with this type of regularizer different latent transformations learn to act on different subspaces and the corresponding representation becomes disentangled. I find this a very elegant approach. The empirical evaluation is sound and I appreciate the varying complexity among the synthetic tasks considered to test individual aspects of the proposed method. This leaves little doubt about how it works in practice. In general, the proposed method overcomes a significant limitation of prior work adopting the symmetry-based notion of disentanglement. Importantly, this method is very accessible and thereby paves the way for others to adopt the framework by Higgins et al. (2018) for learning disentangled representations, which is somewhat under-explored.

Weaknesses: The main weakness of the paper is that the authors do not explore the limits of their approach. The proposed method solves all of the synthetic tasks, which leaves open how it compares on more standard benchmarks for disentanglement. In particular, it seems to me that the applicability of this method is contingent on SO(n) being an adequate group to describe the interactions in the environment at a latent level. There is some evidence that this is not problematic in the experiment with Lie groups, but it would be good to provide additional insight into this through a combination of empirical results on known (more complex) benchmarks and a discussion of limitations in the paper. The final experiment on multi-step prediction is somewhat underwhelming due to the choice of baselines. This is essentially a classic video modeling task (while conditioning on actions) and the authors should compare against a corresponding baseline. For example, an LSTM would already provide a much stronger baseline compared to the “direct prediction baseline”, but ideally a more sophisticated baseline is used, e.g. CDNA (Finn et al., 2016), PredNet (Lotter et al., 2017), SV2P (Babaeizadeh et al., 2017). More generally, this task is somewhat disconnected from the earlier results and I would encourage the authors to replace it (or move it to the appendix) with a more sophisticated disentangled representation learning task.

Correctness: The applied empirical methodology is correct and as far as I can tell the authors do not overstate their claims following the presented results.

Clarity: The paper is well written and easily accessible.

Relation to Prior Work: The authors can improve in this regard by being more specific about the limitations of prior work, and in particular about Caselles-Dupre et al. (2019).

Reproducibility: Yes

Additional Feedback: I would have appreciated a comparison to Caselles-Dupre et al. (2019) on at least one of the synthetic tasks, even though it makes stronger assumptions. In fact, by demonstrating explicitly what kind of assumptions are needed to make the method from Caselles-Dupre et al. (2019) work it becomes clearer how your method improves. The results presented in section 5.4 on the Lie groups are encouraging. However, it also seems to me that when using a neural network to approximate the group homomorphism, it is more susceptible to learn an entangled representation when the number of latent dimensions isn’t exactly 3. Therefore I would like to know what happens if n > 3 in this case (as is explored for the FlatLand task) and encourage the authors to add these results to the appendix. (189) I don’t think that it is reasonable to call CCI-VAE state of the art anno 2020. (304-305) Please also incorporate more recent findings on the usefulness of disentangled representations, such as Locatello, Francesco, et al. "Challenging common assumptions in the unsupervised learning of disentangled representations." international conference on machine learning. 2019. van Steenkiste, Sjoerd, et al. "Are Disentangled Representations Helpful for Abstract Visual Reasoning?." Advances in Neural Information Processing Systems. 2019. -------------------------------- POST REBUTTAL ------------------------------------- I have read the other reviews and the author's response. While the rebuttal did address some of my concerns, I can not raise further. Especially, since I would still like to see an experiment + analysis added on a standard benchmark where the proposed method **fails** (perhaps this is the case for the promised experiments on 3D cars or 3D shapes, but this is not clear from the text). In this way, it should be easier for others to follow-up on this work. I also recognize the scalability issues of the proposed method as pointed out by R2 and R5, which I initially had not considered. I agree that this is an issue that should be discussed in the paper and ideally computational complexity is empirically analyzed. However, considering that the field of disentanglement is still rather nascent and mostly concerned with synthetic datasets and overengineered methods, I don't think this is reason for rejection or a lower score.


Review 3

Summary and Contributions: Motivated by group theory, this paper proposes a new method to learn transformations occurring in sequential data. The method learns to map these transformations to disentangled rotations in the latent space of an auto-encoder. On toy datasets, the method learns representations that correctly capture the group structure of the underlying transformations, unlike prior methods such as CCI-VAE.

Strengths: This theoretically motivated contribution is original, sound and very promising for the field of disentanglement. I enjoyed reading this paper and would like to congratulate the authors for their interesting findings.

Weaknesses: The method is only demonstrated on toy datasets. However, given the quality of the theoretical contribution, the novelty and promises of the method, I still believe it is a strong contribution. The paper could be more highly rated if it adresses the concerns below about clarity and connection to prior work.

Correctness: Yes.

Clarity: Some important details are missing in the main text: 1) training procedure (line 128): is there an inner optimization loop for the thetas and an outer loop for the encoder/decoder? Please clarify the training procedure. It should not be necessary to read the pseudo-code in the suppl material (which I did not find any reference to in the main text) to understand these important aspects of the model. 2) Describe all network architectures in text or supplementary material (how many layers etc). A link to the notebook code complements but does not replace this description. 3) fig2: what do the different symbols/colors correspond to? 4) "5.4 Learning disentangled representations of Lie groups" This part of the paper need more details and explanations. a. Why do the authors need to introduce an alternative architecture for Lie Groups? The theta parametrization in 5.2 and 5.3 is already continuous and should be able to map to a Lie Group. b. I don't understand how the neural network \rho_s fits within the encoder/decoder architecture previously described. 5) part 5.5 a. Missing details on how the g_alpha actions are fed to the model. b. Is the model first pre-trained in an unsupervised way on trajectories, before being used to predict the next frames? c. A figure showing an example of reconstructed trajectory would be useful.

Relation to Prior Work: IMPORTANT (PLEASE ADRESS): Connor and Rozell (AAAI Feb. 2020) seem to have developed a very similar framework in parallel to this work. Although this present study is different in many ways, I think it would be important to see an extended discussion on the similarities and differences of the two approaches in the related work section. The authors mention that one difference with their framework is that Connor and Rozell assume that the transformations are small but: 1) Reading Connor and Rozell's paper, it is not clear to me that their method could not work on larger transformations as well 2) The present study also trains on a sequence of small consecutive transformations, so the difference claimed here between the two studies *does not seem valid to me*.

Reproducibility: Yes

Additional Feedback: EDIT: I am satisfied with the authors' response, but the limited information present in the rebuttal does not allow me to increase my rating with confidence. These comments and questions are intended to enrich the discussion and don't need to be addressed in the rebuttal: 1) It seems that the regularization chosen (low dimensionality of single transformation steps) matches the specific statistics of the toy problem used (l164: the ball moves either left right up or down at each step), but it is unclear to me whether this choice of regularization would generalize to more natural problems. 2) l 119: why use special orthogonal matrices, and not another family (e.g. unitary matrices)? What groups of transformations acting on natural images can this family of matrices represent / not represent? 3) l120: Why constrain the input to be unit normed? There is a non-linear encoder between the input and the latent. Did the authors see an empirical benefit of doing this? 4) Why uses whole trajectories at train time instead of pairs of consecutive data points? Minor: eqn 3: it seems that the same angle theta is applied at each step in this equation. Shouldn't it be a different theta for each step? broader impact: it is quite clear to me that this kind of work could lead to more interpretable latent representations, with implications for the design of robust, reliable and fair ML systems.


Review 4

Summary and Contributions: This paper tackles the problem of learning disentangled representations from a sequence of observed transformations. It is based on the recently proposed notion of disentanglement from the perspective of symmetry preserving group representations. The authors propose to parametrizing transformations in the learned latent representation space with a series of pairwise rotation matrices. In this way they are able to show that the group structure of the given transformations can be accurately recovered and disentanglement can be achieved with a simple sparsity promoting regularizer.

Strengths: The recently proposed group-theoretic notion of disentanglement upon which this work builds is a very promising formalization. However, this paper is among the first to actually propose an method which uses this framework to learn disentangled representations. The proposed method is clearly novel, well founded, intuitively appealing and very relevant to representation learning. The empirical evaluation clearly demonstrates the benefit of this method, and that it is indeed able to learn the correct structure of the relevant transformation groups.

Weaknesses: My main concern with this paper is scalability: The proposed parametrization requires a number of pairwise rotation matrices that scales quadratically in the number of latent factors, which may make it impractical for large latent spaces. Furthermore given that this parametrization is the central contribution of this paper, it is unclear how much of the proposed framework would transfer to more complex environments which may require more expressive group structure than SO(n). A further concern is that the proposed method depends on supervised information about the applied transformation. This limits its applicability to synthetic data, or cases where this data is available (eg. in the form of actions).

Correctness: I have no concerns regarding the correctness of the method or the methodology.

Clarity: The paper is well written and easy to understand.

Relation to Prior Work: Relation to other notions of disentanglement is briefly discussed, but that might suffice. As far as I can tell, not much other related work exists so far, because this approach to disentanglement is relatively novel.

Reproducibility: Yes

Additional Feedback: In line 239-241 it is claimed that additional latent dimensions does not change the learned representation. Given the sparsity regularization, I am inclined to believe that, but nonetheless it would have been nice to see this confirmend in an experiment. ------------------------ POST REBUTTAL ----------------------- After reading the rebuttal I realized that scalability is less of an issue than I originally thought. I was worried that the number of required rotation matrices increased quadratically with the size of the representation. However, I did not take into account that each rotation matrix is inherently very sparse with only 4 learned parameters independent of the dimension (as the authors point out). With that, and the additional evidence that unused dimensions are in fact ignored in practice, I am now convinced that this is a solid paper overall, and raise my score to 7.

[Author Response · NeurIPS 2020]

We thank all reviewers for their comments and for their time in reviewing our work. We appreciate the reviewers' assessment of the strengths of the paper, with all reviewers highlighting our contribution towards putting into practice the symmetry-based notion of disentanglement. We address their questions and issues in the following.

**Limitations and scalability** *(we will amend our paper to discuss the following point)*

Reviewers 2, 3, and 5 note that the discussion of our work's limitations could be improved. Although $SO(n)$ readily occurs in many systems of interest, there are indeed environments where this will not be the case — for example, if objects appear and disappear, or split apart and merge together. With regards to scalability, there is a polynomial scaling associated with the $n(n-1)/2$ rotations that describe a transformation in an n-dimensional spherical latent space. Whilst each of these $n \times n$ rotation matrices is relatively sparse (with 4 parameterised elements as per eq. (2) of the manuscript), ultimately it is unavoidable that computational cost increases with system size.

**Related Work** *(the following discussions, and the papers mentioned by reviewers 2 and 3, will be added to our work)*

Reviewer 4 enquires about the relation between our work and Connor and Rozell (AAAI Feb. 2020), which is indeed insufficiently described in our paper. Their work requires first learning a structured latent space with an auto-encoder, which, as we show in our figure 2, requires neighbouring states to overlap. In their experiments, to satisfy this condition, differences between original states and transformed states had to be made quite small. We speculate that their method requires this first step because their use of general matrices (instead of $SO(n)$) to represent transformations may not enforce a sufficiently strong prior on the type of transformations operating on their system. Another important difference between their work and ours is that their sparsity regularisation aims to reduce the number of transport operators that describe the system and not enforce disentanglement within each operator as in our method.

Reviewer 2 has some questions concerning the connections between our work and both static and dynamical VAEs, which we indeed insufficiently explain. Our work and SSMs use some similar tools, and interesting representations can indeed be found in both. However, there are fundamental differences that lead to us treating them separately and not using SSMs as experimental baselines. First and foremost, it is not straightforward to reconcile the SSM objective of modelling probabilistic generative processes with the inherently deterministic framework of Higgins et al (which is why we do not use a VAE). Our method is the first to propose a path to disentanglement (in the sense of Higgins) in either deterministic or probabilistic frameworks. As such, we feel that attempting to bridge the gap between symmetry-based representation learning and SSMs is certainly an exciting research direction but is outside the scope of our current work.

Reviewer 3 asks a question about the limitations of the work of Caselles-Dupré, which we indeed insufficiently describe. Their method only applies to transformations that are identity except for a single unknown 2x2 block on the diagonal (Sec 6.2 of their paper); their formalism therefore applies to the gridworld experiment, where they do find similar representations to ours, but not to the other environments considered in our work. Importantly, their framework can only work when the type of symmetries are 1) a priori known and 2) described by this type of matrix.

**Experiments** *(results to be added to the appendix)*

Reviewers 3 and 5 have questions about experimental results when we increase the number of latent space dimensions. For the Lie group experiment with a latent space dimension of 4, we do find that the additional dimension is ignored (see figure). Similar results are found for a latent space dimension of 6 for lines 239-241.

Reviewer 3 is correct that the "direct prediction" baseline in Sec 5.5 could be replaced by a stronger baseline. However, the intent of this experiment is simply to demonstrate that the representations learnt by our framework do indeed exhibit desirable properties (beyond interpretability) that are often associated with disentanglement. The chosen baselines are then intended as direct ablations of our regularisation, and (in the case of "direct prediction" ) our prior assumptions on the environmental structure. With regards to reviewer 3's suggestion that we present a more complex disentanglement task, we did apply our framework to the established datasets of 3D Cars (Kim and Mnih, "Disentangling by factorising", 2019) and 3D Shapes (Reed et al. "Deep visual analogy-making", 2015). We ultimately decided that the experiments in the main text sufficiently presented our contribution, but would be happy to include these results in the appendix.

**Clarifications** *(the following clarifications will be added to our paper)*

To address Reviewer 4's specific queries: 1) The encoder, decoder and thetas are all learned within the same loop. 2) We will fully describe all network architectures in supplementary Sec 3.3. 3) Each symbol/colour pair corresponds to a unique state of the environment. This mapping is consistent for all latent spaces. 4) In Sec 5.2 and 5.3, independent sets of parameters (thetas) are learnt for each discrete environmental actions. In Sec 5.4, this look-up-table is replaced by a network, $\rho_\sigma$, that maps (continuous) environmental actions to rotations in the latent space. 5) The same training protocol as for the other experiments was used here too; further information will be added to the appendix.

[Meta-Review · NeurIPS 2020]

4 knowledgeable reviewers reviewed the paper. Requested clarifications and points raised by the reviewers, s.a. initially missed related work discussions, were adequately addressed in the rebuttal. As a result two reviewers increased their scores, 3 voting to accept the paper, while R2 increased his score by 1 still judging the paper marginally below threshold and requiring further work. The main weakness remains the limited empirical validation, limited to toyish data, and in comparison to a limited range of baselines. All reviewers however agreed that the paper provides a novel, original and interesting contribution to the topic of disentangled representations learning. Thus, despite the mentioned limitations, and in agreement with the majority of reviewers, the AC recommends to accept the paper. We expect the clarifications form the authors' response to be integrated in the final paper. Also authors should include a more thorough discussion of scalability considerations of the method, and an explicit discussion of its limitations (as requested by R2).